# The Safety and Efficacy of Psychosocial Adherence Interventions in Young People with Early Psychosis: A Systematic Review

**DOI:** 10.3390/healthcare10091732

**Published:** 2022-09-09

**Authors:** Gül Dikeç, Ellie Brown, Daniel Bressington, Andrew Thompson, Richard Gray

**Affiliations:** 1Department of Nursing, Faculty of Health Sciences, Fenerbahçe University, Istanbul 34758, Turkey; 2School of Nursing and Midwifery, La Trobe University, Bundoora, Melbourne, VIC 3086, Australia; 3Centre for Youth Mental Health, Faculty of Medicine, Dentistry and Health Sciences, The University of Melbourne, Melbourne, VIC 3052, Australia; 4College of Nursing and Midwifery, Charles Darwin University, Darwin, VIC 0810, Australia

**Keywords:** adherence, early psychosis, psychosocial interventions, systematic review

## Abstract

Background: The role of antipsychotic medication in supporting young people in their recovery from early psychosis is complex and controversial. It is common for young people, often given antipsychotic medication for the first time, to express a choice to stop treatment, potentially increasing the risk of relapse and admission to hospital. Our systematic review aimed to evaluate the safety and effectiveness of psychosocial interventions to enhance antipsychotic medication adherence in young people with early psychosis. Methods: We reviewed studies using any experimental design of psychosocial interventions specifically focused on enhancing adherence with antipsychotic medication in young people with early psychosis. Cochrane CENTRAL Register, Medline, Embase, PsychINFO and CINAHL were searched on 19 November 2021 without time restriction. Studies were assessed for quality using the Effective Public Health Practice Project Quality Assessment Tool for Quantitative Studies. Results: Our initial search identified 3469 documents. Following title, abstract and full-text screening, we included three published studies and one unpublished experimental study that met our inclusion criteria. Outcome data were available for three studies that tested adherence–coping–education, adherence therapy, and a health dialogue intervention, all having a positive effect on medication adherence. None of the trials reported data on the safety of the experimental interventions. Conclusion: There is a paucity of evidence from high-quality randomized controlled trials that establish the safety and effectiveness of any type of psychosocial intervention to enhance medication adherence in young people with early psychosis. Further high-quality trials are warranted. This review was registered on the Open Science Framework prior to undertaking out initial searches.

## 1. Introduction

Treatment with antipsychotic medication is the standard of care for patients with early psychosis to both manage symptoms and prevent symptom relapse [1,2]. The relationship between antipsychotic medication and service user’s recovery and wellbeing is complex; antipsychotic medication is often considered, by service users, as an intervention that is imposed on them by clinicians, and their personal agency is denied. The decision to not take medication is clinically conceptualized as non-adherence but may equally be constructed as a choice to manage symptoms without medication. Authors have reported rates of stopping medication in young people with psychosis of around 50% in the year after starting antipsychotic treatment [3,4,5], increasing to 75% in the second year [6]. A cohort study of 605 patients with early psychosis reported that 19% refused antipsychotic treatment persistently over the 18-month study period [3].

Alvarez-Jimenez et al. [7] reported a significant association between lower adherence and relapse in a systematic review and meta-analysis of 29 studies involving 3978 young people with psychosis. The meta-analysis suggested that non-adherence with antipsychotic medication led to a four-fold increase in the odds of relapse (OR = 4.09, 95% CI 2.55, 6.56; *p* < 0.01]. This observation has been confirmed in subsequent studies, although the size of the effect was more modest. For example, in a group of 136 young people, the odds of relapse increased by 50% over the one-year study period if participants stopped taking their medication [2]. 

The reasons why people do not take medication as prescribed are complex [7,8,9] and may vary between patients and within the same individual over time. Perhaps the most important factors affecting adherence in young people with early psychosis that are reported in the literature are the duration of untreated symptoms [10,11], substance use (addiction) [3,11], lack of insight (not accepting the symptoms as part of mental ill-health) [4,12], medication-related side effects [3,13] and lack of social support [4,14].

Refusal to take antipsychotic medication in people with psychosis is often addressed using long-acting injections (LAIs or depots) of medications rather than through psychosocial intervention [15]. Whilst there is good evidence that LAIs are more effective at preventing relapse than oral medication, presumably because patients are more adherent to treatment, the intervention denies people the opportunity for meaningful discussion and reflection about the objective and subjective effects of medication [5]. However, some authors argue that LAIs are under-prescribed in people with psychosis [16]. Further, psychiatrists can be reticent to prescribe long-acting forms of medication to this group of people [17].

Given the apparent reluctance to address non-adherence using LAIs, psychosocial interventions to enhance adherence with oral antipsychotic medication may be an important part of a care package for people with psychosis. For example, a systematic review and meta-analysis by Gray et al. [18] showed that adherence therapy (AT), a structured psychosocial intervention based on motivational interviewing and CBT and with a focus on shared decision making, was effective at improving psychiatric symptoms with a medium effect size (*g* = −0.56, 95% CI −1.03, −0.09; 707 participants). Adherence therapy is built around five core interventions: (1) detailed assessment of medication beliefs and side effects; (2) an exploration of past experiences with medication, (3) solving problems with medication (e.g., forgetting to take medicine, side effects), (4) exploring beliefs about medication; and (5) examining how medication will help or hinder future plans [18]. Of the six studies included in this systematic review, none focused on people experiencing a first episode of psychosis. 

Treatment with antipsychotic medication remains the standard of care for many young people with early psychosis [19]. Psychosocial adherence interventions may be a helpful adjunct treatment to enhance medication adherence but have not been systematically reviewed. This systematic review aimed to evaluate the effectiveness of psychosocial interventions to enhance adherence with antipsychotic medication in young people with early psychosis.

## 2. Methods

### 2.1. Review Design

The Preferred Reporting Items for Systematic Reviews and Meta-Analyses (PRISMA) updated guidelines were followed in reporting this systematic review [20]. The research team comprised nurses, epidemiologists, psychiatrists, and psychologists. We aimed to systematically review all experimental trials testing a psychosocial intervention focused on enhancing medication adherence in young people aged 15 to 25 years experiencing early psychosis. The primary outcome of interest was the adherence to treatment, determined using any standardized measure. The research question was therefore to establish the safety and effectiveness of psychosocial interventions to improve adherence with antipsychotic medications in young people with early psychosis. 

### 2.2. Pre-Registration

The protocol for this review was prospectively registered by the Open Science Framework (OSF) registry on 12 November 2021, with registration https://doi.org/10.17605/OSF.IO/B3EXZ (accessed on 12 November 2021).

### 2.3. Inclusion Criteria

We included experimental studies where the following PICO (population, intervention, control, outcome) criteria applied: Population: young people (aged between 15–25 years) experiencing first-episode psychosis (defined as within the first year of treatment by mental health services).Intervention: any psychosocial intervention (where the focus was on enhancing medication adherence), delivered by any healthcare worker (e.g., psychologist, nurse) in any clinical setting (e.g., early psychosis service, primary care), via any medium (e.g., telephone, face-to-face).Comparator: any (e.g., treatment as usual, attentional control).Outcomes: medication adherence determined using any routine measure including, but not limited to, the drug attitude inventory, the Medication Adherence Rating Scale, pill count.

Additionally, the trials needed to be written in English. If the authors reported an extension or secondary analysis of a previously reported trial, only manuscripts reporting primary data were included; this was done to eliminate the risk of counting the same participants twice in the same review.

### 2.4. Data Sources and Search Strategy

A systematic search of the literature for relevant articles published from database inception until November 2021 was undertaken. We searched the following databases: PsychINFO (using Ovid), Medline, Embase, PubMed, CINAHL and CENTRAL (Cochrane Central Register of Controlled Trials). Our initial search strategy was developed in MEDLINE and was then adapted to other databases (Appendix A). The search was conducted on 19 November 2021.

The web-based systematic review management software COVIDENCE (www.covidence.org, accessed on 28 July 2022) was used for this study. COVIDENCE is a fully auditable package used for title, abstract and full-text screening; data extraction; and quality appraisal. At least two researchers completed title, abstract and full-text screening (G.D., R.G., E.B. and D.B.) independently. Discrepancies were resolved by discussion between investigators or consultation with a third member of the team. We also checked the reference lists of the included studies to identify additional relevant trials. The grey literature was not searched as these documents had not been subject to peer review. 

### 2.5. Data Extraction and Qualitative Synthesis

The following data were extracted again by two researchers independently with any discrepancies being resolved by discussion (G.D. and R.G.). The following data were extracted (also using COVIDENCE) from included studies: author, the country where fieldwork was undertaken, trial registration status, clinical setting, intervention, duration of intervention, comparator intervention, duration of comparator intervention, mode of delivery, primary endpoint, number of participants randomized, number of participants used in the analysis and adherence outcomes. We undertook a qualitative synthesis and intended to conduct a meta-analysis using Review Manager Version 5.

### 2.6. Assessment of Methodological Quality

Study quality was determined using the Effective Public Health Practice Project (EPHPP) Quality Assessment Tool for Quantitative Studies [21], which has been extensively used in systematic reviews and meta-analyses. Bias was assessed in six domains: (1) selection bias; (2) study design; (3) confounders; (4) blinding; (5) data collection method; and (6) withdrawal/dropouts. Each domain was rated as strong (3 points), moderate (2 points) or weak (1 point). Based on the total score, studies are assigned a quality rating of weak, moderate, or strong [21]. Two authors (R.G. and G.D.) independently rated each study for risk of bias and subsequently discussed their ratings to reach a consensus.

## 3. Results

### 3.1. Search Results

Figure 1 is a PRISMA diagram showing the flow of papers through the study. Our initial search identified 3469. Three published and one unpublished trial (a total of four manuscripts) met our study inclusion and exclusion criteria and were included in our review.

### 3.2. Characteristics of Included Trials

Table 1 and Table 2 show the characteristics of and data extracted from included trials.

### 3.3. Included Studies

Three included studies had considerable overlap in terms of a therapeutic approach. Two of them used a form of AT among young people with EP [8,22]. One of them [23] was a health dialogue program that combined CBT and psychoeducation, similar to AT. One trial tested a smart-phone application [24]. All studies were carried out in outpatient clinics, two studies were conducted in the USA [22,23] and one each in the UK [8] and China [25].

Uzenoff et al. [22] reported a pilot randomized controlled trial (RCT) to examine the effectiveness of adherence–coping–education (ACE) Therapy compared with supportive treatment (ST). ACE was a new adaptation of compliance/adherence therapy [18,26,27], although the duration of ACE was longer than that of AT (14 sessions vs. 8 sessions, respectively) [18]. The trial included twenty-four patients with first-episode psychosis. Assessments were conducted at baseline, mid- (3 months) and post-treatment (6 months). Adherence was determined using patient self-reporting and treatment attitudes. A difference in treatment attitudes was observed at both the mid- and post-treatment assessments. Safety data were not reported.

A mirror image study evaluating the effectiveness of AT training for two multidisciplinary early intervention in psychosis (EIIP) teams was reported by Brown et al. [8]. AT training involved 6 days of training delivered over a 6-month period. The primary outcome of the study was the differences in relapse rates (a proxy adherence measures) in the year preceding and the year after training. Relapses reduced from 20 in the year before to 9 in the year following training, a reduction that equates to a medium effect size (0.33). No safety data were reported.

Weiden et al. [23] report an RCT comparing a health dialogue adherence intervention (based on principals of the INSIGHT CBT model) [24] with psychoeducation in 34 patients with first-episode psychosis. Participants received between 5 and 22 sessions over a six- to twelve-month period. Adherence was determined using the “all-source verification” method that summarizes adherence information from multiple sources into a weekly adherent/not adherent composite rating. The primary outcome was the time until the participant completely stopped antipsychotic medication for more than a week. Participants in the health dialogue group continued with treatment for an average of 47 weeks compared to 23 weeks in the control group. Safety data were not reported in the manuscript. Only a conference abstract was available for this trial; we contacted the authors on the 17 March 2022 to ascertain if additional information was available. The authors did not respond to us.

### 3.4. Unpublished Trial

We identified one unpublished trial that was prospectively registered with the Chinese Clinical Trial Registry in 2018 (ChiCTR1800017286) [24]. The experimental intervention was a smartphone application that included medication reminders, outpatient prompts, side effect monitoring and psychoeducational information. Participants were asked to use the application for 12 weeks. The control group participants were offered a placebo intervention that was not described in the registration entry. The authors aimed to recruit 100 patients. The primary outcome was medication adherence determined using the Medication Adherence Rating Scale (MARS) [28], Drug Attitudes Inventory (DAI) [29] and pill count at the end of treatment (12 weeks) [24].

According to the registry entry, the trial was completed on the 31 December 2019. We contacted the authors (on 16 March 2022) to inquire about their data availability. They reported that the study was finished, but it currently has not been published in a peer-reviewed journal, and the authors did not provide us with any data. We checked on 17 March 2022 using the trial registration number to determine if the results had been published; they had not.

### 3.5. Quality of Trials

Table 3 shows the critical appraisal for the included studies. Based on the available information, we rated the Uzenoff et al. [22] trial as strong and the Brown et al. [8] and Weiden et al. [23] trials as weak. We did not complete the risk of bias rating on the Wong [24] trial because the results have yet to be published.

## 4. Discussion

This systematic review aimed to evaluate the safety and efficacy of psychosocial interventions to enhance adherence to antipsychotic medication in people with early psychosis. Our search identified two included studies that were pilot studies [22,23] and a mirror image study [8] that met our inclusion criteria, of which all reported outcome data [24]. We were not able to extract data from the fourth study, which was uncompleted [24]

The interventional approach adopted across the three trials where data were available was broadly similar in that the focus was on approaches drawn from CBT and motivational interviewing. Across the studies, there was an apparently positive impact on medication adherence, although we note the methodological quality of the included studies varied considerably. We note that harms were not reported across all the included studies.

Our observation contrasts with other reviews of adherence interventions in the general adult psychiatric population that have tested different psychosocial interventions to enhance adherence [30,31]. For example, El Abdellati et al. [32] reviewed 17 trials of adherence interventions in adult patients with psychosis. They reported consistent evidence that interventions involving family, technology (smartphone reminders), psychoeducation, CBT and AT were effective at enhancing adherence [32]. However, it may not be appropriate to generalize evidence from an adult population to young people experiencing a first episode of psychosis because they are populations that differ in many important ways. For example, young people are more susceptible to side effects from medication (e.g., weight gain and sexual dysfunction) than adults with psychosis [18,33].

One possible reason for our finding of a lack of trials investigating medication adherence in first-episode psychosis is that maintaining medication after recovery from a first episode remains a relatively contentious issue. Young people may also express a choice for care and treatment without medication. That said, for a substantive proportion of young people, medication continues to be an important part of treatment, and adherence is required to prevent the occurrence of functionally and economically costly relapses.

The use of technology to enhance medication adherence, such as the application of short messaging service (SMS) prompts, can be effective [34,35]. It may be that further research on the application of technology-based adherence interventions in this population is justified.

### Limitations of This Review

There were important limitations to our review that require consideration. We did not co-produce this review with a young person with lived experiences of taking antipsychotic medication; on reflection, this was an important omission that would have enhanced the relevance of our review.

The outcome of this review was medication adherence. Although this may be important for clinicians, we suspect that service users may not consider this a relevant focus.

We included one study where adherence was determined using relapse (admission to hospital). It is debatable if relapse is a valid adherence measure [11], and it could be argued that the trials conducted by Brown et al. [8] should have been excluded from this review.

We identified several trials of cognitive adaptation training (CAT) that we excluded from our review because the interventions were not exclusively or primarily focused on improving treatment adherence; rather, they focused on the community adaptation of patients with psychosis. We note that in other systematic reviews of adherence interventions in adults with psychosis, e.g., El Abdellati et al. [32], CAT trials have been included as an adherence intervention. It could be considered that our exclusive definition of an adherence intervention was a limitation of this review.

Medication adherence may be enhanced by psychosocial interventions delivered as standard by early intervention services. As these are assumed to have been already successfully incorporated into the treatment packages young people receive, they have not warranted evaluation as discrete interventions. For example, Garety et al. [36] reported an RCT of specialized care for early psychosis that resulted in significant improvements in medication adherence, in addition to satisfaction, functioning and quality of life over the 18-month study period.

## 5. Conclusions

This systematic review aimed to synthesize evidence from trials about psychosocial interventions to improve medication adherence in young people with early psychosis. Due to a paucity of trials, we were unable to determine if psychosocial interventions were safe or effective. The language of adherence is perhaps discordant with recovery principals, and we have reflected during the conduct of this review on the importance of the words we choose when talking to service users about medication [37]. That said, antipsychotic medication is often important in helping young people with early psychosis to manage the symptoms they are experiencing. In the absence of evidence, clinicians need to focus on agency and choice. There also needs to be acceptance and support by clinicians of the decisions made by service users about their treatment.

## Figures and Tables

**Figure 1 healthcare-10-01732-f001:**
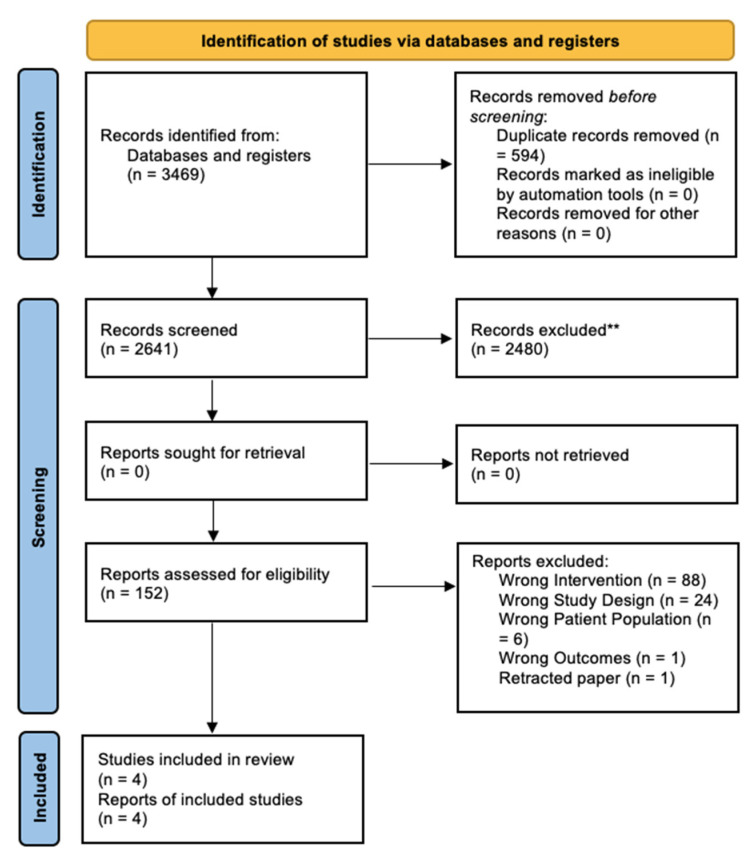
Flow of documents through the review process.

**Table 1 healthcare-10-01732-t001:** Characteristics of included studies.

Author	Country Where Fieldwork Was Conducted	Clinical Setting	Intervention	Duration of Intervention	Adherence Measures	Primary Endpoint
Uzenoff et al. [22]	United States of America (USA)	Psychiatric Inpatient and Outpatient Clinics	Adherence–Coping–Education Therapy (ACE) vs. Supportive Therapy (ST)	14 sessions over a six-months period	Patient Reports; “Need for Treatment” “Benefits of Medication “Variables in Rating of Medication Influences Scale and Insight and Treatment Attitudes Questionnaire	Adherence at the end of treatment
Brown et al. [8]	United Kingdom (UK)	Early Intervention in Psychosis (EIIP)	Adherence Therapy Training to EIIP Teams	6 1-day monthly in 6-months	Relapse Rates	Relapse over twelve months from start of study
Weiden et al. [23]	United States of America (USA)	Psychosis Disorders Program	Health Dialogue Intervention	6 to 12 months	All Source Verification (ASV)	Twelve months
Wong [25]	China	Psychiatric Outpatient Clinic	Smartphone Application	3 months	Medication Adherence Rating Scale (MARS); Drug Attitude Inventory (DAI); Pill Count	Three months

**Table 2 healthcare-10-01732-t002:** Extracted data from included studies.

Author	Trial Registration Status	Number of Participants Randomized	Number of Participants Included in the Analysis	Adherence Outcomes at the End of the Trial
Uzenoff et al. [22]	Not reported	*n* = 24	*n* = 19	ACE participants on benefits of medication scores (*d =* 0.59) were significantly improved in post-treatment.
Brown et al. [8]	Not reported	Mirror Image Study (N/A)	Patient *n* = 35	Statistically significant reduction in relapses, equating to a medium effect size (0.33 [95% CI = 1.13–2.66]).
Weiden et al. [23]	Not reported	*n* = 34	*n* = 34	Experimental group stayed on medication longer than PE subjects. (46.7 weeks [95% CI 27.3–66.1] compared to 22.5 [95%CI 9.6–35.5])
Wong [24]	Prospectively registered (ChiCTR1800017286)	*n* = 100	Not reported	Not reported

**Table 3 healthcare-10-01732-t003:** Quality assessment using EPHPP.

	Selection Bias	Study Design	Confounders	Blinding	Data Collection Method	Withdrawals/Dropouts	Global Rating
Uzenoff et al. [22]	Strong	Strong	Strong	Moderate	Strong	Strong	Strong
Brown et al. [8]	Strong	Weak	Weak	Moderate	Strong	Moderate	Weak
Weiden et al. [23]	Moderate	Strong	Weak	Moderate	Moderate	Weak	Weak
Wong [24]	-	-	-	-	-	-	-

## Data Availability

No new data were created or analyzed in this study. Data sharing is not applicable to this article.

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
