# Peer review of "The Safety and Efficacy of Psychosocial Adherence Interventions in Young People with Early Psychosis: A Systematic Review"

_healthcare, 2022, doi:10.3390/healthcare10091732_

Round 1

Reviewer 1 Report

After reviewing the manuscript, I would like to comment that the statistical analysis should be explained more briefly. The prevention of a relapse also depends on the antipsychotic drug, which has been chosen. Maybe it should be worth mentioning to examine the SNP of certain risk genes to better chose the antipsychotic drug. However, this is an important research article.

Author Response

Reviewer I Comments

Authors’ Response

After reviewing the manuscript, I would like to comment that the statistical analysis should be explained more briefly.

Thank you for your contributions.

The prevention of a relapse also depends on the antipsychotic drug, which has been chosen. Maybe it should be worth mentioning to examine the SNP of certain risk genes to better chose the antipsychotic drug.

Thank you for your recommendation. Since the manuscript was related to adherence among young people with early psychosis, we believe that it was not about SNP directly. 

However, this is an important research article.

Thank you for your contributions.

Reviewer 2 Report

Dear authors

The article entitled “The Safety and Efficacy of Psychosocial Adherence Interventions in Young People with Early Psychosis: A Systematic Review” is interesting and highlights the need of monitoring medication adherence in young patients having phycological problems. Although very limited, only 3 articles have been selected for the analysis, it is understandable because of the lack of trials and precise information available about this particular issue. Therefore, it is highly recommended to highlight the significance of these types of studies in this manuscript. What type of parameters should be considered and what do you recommend for such types of study design in the future? Is there any data available that shows the involvement of technology e.g, any particular application that helps in improving medicinal adherence in young people, or is there any need to develop some technology in this regard except having simple reminders on a smartphone? The discussion part should be expanded by adding the significance of having such trials and recommendations for future studies.  

The authors have analyzed the data properly, however, the manuscript should go under general refinement by adding some information mentioned/highlighted above and correcting some typo mistakes mentioned in the manuscript. 

Author Response

Reviewer II Comments

Authors’ Response

The article entitled “The Safety and Efficacy of Psychosocial Adherence Interventions in Young People with Early Psychosis: A Systematic Review” is interesting and highlights the need of monitoring medication adherence in young patients having phycological problems. Although very limited, only 3 articles have been selected for the analysis, it is understandable because of the lack of trials and precise information available about this particular issue. Therefore, it is highly recommended to highlight the significance of these types of studies in this manuscript.

What type of parameters should be considered and what do you recommend for such types of study design in the future?

Is there any data available that shows the involvement of technology e.g, any particular application that helps in improving medicinal adherence in young people, or is there any need to develop some technology in this regard except having simple reminders on a smartphone?

The discussion part should be expanded by adding the significance of having such trials and recommendations for future studies.  

Thank you for your contributions.

We added and some information about technology-based interventions (lines 316-319).

The authors have analyzed the data properly, however, the manuscript should go under general refinement by adding some information mentioned/highlighted above and correcting some typo mistakes mentioned in the manuscript. 

Thank you for your recommendations and some corrections throughout the manuscript.
